# Avermectin B1a Shows Potential Anti-Proliferative and Anticancer Effects in HCT-116 Cells via Enhancing the Stability of Microtubules

**Qendresa Hoti, Duygu Gencalp Rustem and Ozlem Dalmizrak ***

Department of Medical Biochemistry, Faculty of Medicine, Near East University, TRNC, Mersin 10, Nicosia 99138, Turkey; qendresahoti1@gmail.com (Q.H.); duygu.rustem@neu.edu.tr (D.G.R.)
* Correspondence: ozlem.dalmizrak@neu.edu.tr

**Abstract:** Avermectins are a group of macrocyclic lactones that are commonly used as pesticides to treat pests and parasitic worms. Some members of the avermectin family, such as ivermectin, have been found to exhibit anti-proliferative activity toward cancer cells. This study aimed to investigate the potential anti-cancer activities of avermectin B1a using the HCT-116 colon cancer cell line. The MTT assay was used to calculate the $IC_{50}$ by incubating cells with increasing doses of avermectin B1a for 24, 48, and 72 h. Flow cytometry was used to evaluate apoptosis following the 24 h incubation of cells. The migration capacity of the HCT-116 cells in the absence or presence of avermectin B1a was also investigated. Finally, tubulin polymerization in the presence of avermectin B1a was evaluated. Avermectin B1a presented anti-proliferative activity with an $IC_{50}$ value of 30 μM. Avermectin B1a was found to promote tubulin polymerization at 30 μM. In addition, avermectin B1a induced apoptosis in HCT-116 cells and substantially diminished their ability to migrate. Avermectin B1a exhibits significant anti-cancer activity and enhances tubulin polymerization, suggesting that it can be used as a promising microtubule-targeting agent for the development of future anti-cancer drugs.

**Keywords:** avermectin B1a; tubulin polymerization; HCT-116 cell line; apoptosis

## 1. Introduction

Avermectins are a group of macrocyclic lactones that are a natural fermentation product of *Streptomyces avermitilis*, a soil microorganism. Avermectins are a class of medicines with numerous therapeutic applications. They are commonly used as pesticides and exhibit extraordinary potency and a broad range of activity against arthropod and nematode parasites [1]. Avermectins are composed of four major components (A1a, A2a, B1a, and B2a), constituting more than 80% of their total composition, as well as four minor components (A1b, A2b, B1b, and B2b), which constitute less than 20% of their total composition. Among these eight components, it has been reported that avermectin B1a exhibits the lowest toxicity in animals and human beings [2] while also demonstrating the most potent insecticidal activity [2,3]. Abamectin is also referred to as avermectin B1 because it is primarily composed of more than 80% avermectin B1a and about 20% avermectin B1b [4,5]. Avermectin B1a, a macrocyclic lactone, is one of the most frequently utilized avermectin family compounds, and it is used in both medicinal and agricultural applications [6].

The globular protein tubulin, in conjunction with microtubule-binding proteins, produces microtubules [7], which are one of the key components of eukaryotic cells' cytoskeletons. Microtubules are formed via the assembly of tubulin molecules, which join together to form hollow tubes consisting of 13 chains of α–β-tubulin dimers called protofilaments [8]. Microtubules are essential for various cellular processes such as mitosis, intercellular

transport, cell motility, and the overall maintenance of cell morphology [9]. In the process of tubulin polymerization, monomers are either removed or added depending on the concentration of tubulin dimers and subunits [10]. If the dimer concentration exceeds the critical concentration, microtubule polymerization takes place, leading to growth; if this concentration falls below the critical level, the microtubule shrinks [11]. Microtubules are considered good targets for anti-cancer drugs since they are dynamic, and their inhibition can prevent cell division at the spindle checkpoint [12]. Several microtubule-targeting agents have been used for cancer treatment [13]. Microtubule-targeting agents disrupt the dynamic behavior of microtubules by directly interacting with either tubulin dimers or microtubules [14]. Microtubule-targeting agents are split into two classes based on how they affect microtubules: the microtubule-polymerizing or -stabilizing agents and the microtubule-depolymerizing agents [15]. Paclitaxel, docetaxel, and epothilones induce the polymerization of microtubules, while colchicine, vinca alkaloids, dolastatin, combretastatin, and 2-methoxyestradiol are known to depolymerize microtubules [16]. By activating the spindle assembly checkpoint, a delay in chromosome congression prevents the metaphase-to-anaphase transition and arrests cells at mitosis [15]. An extended mitotic blockage triggers apoptosis, which results in cell death. Consequently, a promising strategy for treating cancer is to target the microtubules of malignant cells that are actively proliferating [17]. Therefore, drugs that attempt to interfere with the dynamics of microtubules have the potential to serve as efficient anticancer agents [12].

Previous studies have shown that ivermectin, a derivative of avermectin B1 [18], binds and stabilizes parasitic microtubules [19]. Ivermectin also binds to mammalian tubulin, which can lead to mitotic arrest via changing the tubulin polymerization equilibrium [20]. As a result of the review of the current literature, there is no published research that has shown a direct binding relationship between avermectin B1a and tubulin.

Colon cancer induces mortality and morbidity worldwide, and it is one of the most prevalent human malignancies in affluent countries [21]. It is the third most frequent type of cancer and the second major cause of cancer-related death worldwide, accounting for around 1,400,000 new reported cases and 700,000 fatalities. As a result, innovative antitumor therapy for colon cancer is critical [22]. Colorectal cancer can be classified into five stages based on the extent of local invasion, metastasis, and lymph node involvement, with the final stage being the most advanced and having the worst prognosis [23]. The standard approach for treating colon cancer involves chemotherapy, surgery, and radiotherapy, which are tailored to the specific type and stage of the disease. Surgery is the primary treatment method for most cases of colon cancer, while chemotherapy is considered the most effective treatment [24]. However, chemotherapy can have severe side effects that significantly decrease a patient's quality of life. Therefore, researchers are investigating new drugs that target specific areas, some of which have shown promising results when used in combination with chemotherapy [25].

The aim of this study was to investigate the potential microtubule-stabilizing activity of avermectin B1a as well as its anti-cancer properties, including anti-proliferative, apoptotic, and cell migration properties. This study is the first to examine the effect of avermectin B1a on HCT-116 colon cancer cells.

## 2. Materials and Methods

### 2.1. Materials

#### 2.1.1. Chemicals

Avermectin B1a was purchased from BioAustralis (BIA-A1010, BioAustralis, New South Wales, Australia), paclitaxel was obtained from Cytoskeleton (TXD01, Cytoskeleton, Denver, CO, USA), and colchicine was provided by Sigma Aldrich (C9754, Sigma Aldrich, St. Louis, MO, USA). Dulbecco's Modified Eagle Medium (DMEM), the buffer Dulbecco's PBS, and L-Glutamine (200 mM) were purchased from Capricorn (Capricorn, Dusseldorf, Germany), while fetal bovine serum (FBS), penicillin/streptomycin, and trypsin/EDTA

were obtained from Biological Industries (Biological Industries, Jezreel Valley, Israel). The cell-counting dye trypan blue (0.4%) was purchased from Thermo Fisher Scientific (15250-061, Thermo Fisher Scientific, Waltham, MA, USA). Dimethyl sulfoxide (DMSO) was obtained from Carlo Erba (445106, Carlo Erba, Val de Reuil, Normandie, France).

3-(4,5-dimethylthiazol-2-yl)-2,5-diphenyltetrazolium bromide (MTT) was used for cell viability assay and was purchased from Thermo Fisher Scientific (M6494, Thermo Fisher Scientific, USA). A tubulin polymerization assay was performed by using a tubulin polymerization assay kit from Cytoskeleton (BK004P, Cytoskeleton, Denver, CO, USA), whereas the apoptosis assay was performed by using annexin V-FITC/PI apoptosis kit purchased from Elabscience (E-CK-A211, Elabscience, Huston, TX, USA).

### 2.1.2. Preparation of Solutions

All compounds were initially dissolved in 100% DMSO before their further dilution in a cell culture medium. Final DMSO concentrations were kept constant at 0.1% for all samples, including the controls. MTT working solution was separately prepared in 5 mg/mL of PBS. The test compound avermectin B1a and the controls, paclitaxel, and colchicine were dissolved in DMSO.

### *2.2. Methods*

### 2.2.1. Cell Culture

The colon cancer cell line HCT-116 was obtained from American Cell Culture Collection (CCL-247, ATCC). The cells were cultured in DMEM supplemented with 10% FBS and 1% 200 mM L-glutamine and 1% penicillin/streptomycin and cultured at 37 °C in a 5% $CO_2$ atmosphere.

### 2.2.2. Cell Viability Assay

The 3-(4,5-dimethylthiazol-2-yl)-2,5-diphenyltetrazolium bromide (MTT) assay was performed to investigate the effects of avermectin B1a on HCT-116 cell viability. $5 \times 10^3$ cells/mL were seeded into 96-well plates in a 100 μL medium and incubated at 37 °C in a 5% $CO_2$ incubator for 24 h. Cells were then treated with different concentrations (2.5, 5, 10, 15, 20, and 30 μM) of avermectin B1a dissolved in DMSO. After treatment for 24, 48, and 72 h, 10 μL of MTT solution was added to each well. Cells were incubated for 3 h at 37 °C; then, 100 μL of DMSO was added to each well. Plates were incubated for 15 min at room temperature, and absorbance was measured at 570 nm using a microplate reader (Versa Max, Molecular Device, Sunnyvale, CA, USA). The cell viability percentage was determined using the following formula:

$$\text{Viable cells \%} = \frac{Absorbances\ treated - Absorbance\ blank}{Absorbance\ untreated - Absorbance\ blank} \times 100$$

### 2.2.3. Polymerization Assay of Mammalian Tubulin

The Tubulin Polymerization HTS Assay Kit (BK004P, Cytoskeleton, Denver, CO, USA) was used to perform this assay in accordance with corresponding kit protocol. In brief, tubulin was resuspended at a concentration of 4 mg/mL in ice-cold G-PEM buffer (80 mM PIPES, 2 mM $MgCl_2$, 0.5 mM EGTA, 1 mM GTP, and 5% glycerol). The assay was conducted in a pre-warmed 96-well plate in duplicate. Excluding the control, 4 mg/mL of tubulin was added to each well. The compounds were divided into groups: a control group; a group corresponding to the investigated compound, avermectin B1a, at a concentration of 30 μM; controls with a final concentration of 10 μM for each dose of paclitaxel as an enhancer (positive polymerization control) and colchicine as an inhibitor (negative polymerization control) of tubulin polymerization; and DMSO. Light scattering was measured immediately after the addition of tubulin using a microplate reader (Versa Max, Molecular Device, Sunnyvale, CA, USA). We determined microtubule depolymerization and

polymerization kinetic curves by measuring the optical density values at 340 nm at 30 s intervals for one hour.

### 2.2.4. Cell Apoptosis and Flow Cytometry Analysis

The Annexin V-FITC/PI Apoptosis Kit (E-CK-A211, Elabscience, Houston, TX, USA) was used to evaluate apoptosis. In brief, HCT-116 cells were subcultured at a density of 5 × 10^5 cells/mL. The medium was removed 24 h after incubation, and cells were treated with 30 μM avermectin B1a, DMSO, and two separate control groups (the untreated control unstained AN-/PI- and the untreated control stained An+/PI+), with three repetitions conducted for each group. The cells were then incubated at 37 °C for 24 h in a 5% $CO_2$ incubator. The assay was performed according to the protocol provided with the kit. A FACSCalibur flow cytometer (Becton Dickinson, San Jose, CA, USA) was used, in accordance with the instructions provided with the apoptosis detection kit, to investigate cell apoptosis.

### 2.2.5. Wound-Healing Assay

HCT-116 cells were seeded in 6-well plates at 6 × 10^4 cells/well and incubated at 37 °C in a 5% $CO_2$ incubator until they reached over 80% confluency. Then, the HCT-116 cells were incubated with a serum-free medium for 24 h. After the incubation period, 10 μL sterile pipette tips were used to scratch the cell monolayers manually. After scratching, the serum-free medium was replaced with PBS, and each scratched area was considered as the zero-time point (0 h). Photographs of the wound areas were taken at 0 h using an Olympus IX53 Inverted Microscope with a 10X objective. After taking pictures of each group at the respective zero-time points, the PBS was removed, and the cell culture medium was added to each well carefully. HCT-116 cells were treated with avermectin B1a (30 μM) prepared in a cell culture medium for 24 h. As soon as the incubation period had elapsed, the cell culture medium was removed, and the cells were washed with PBS carefully. The scratched areas were re-photographed using an Olympus IX53 Inverted Microscope. The experiments were performed in quadruplicate, and DMSO was used as a control. ImageJ software was used to quantify the pictures, and wound closure percentage was calculated using the following formula:

$$\text{Wound Closure \%} = \left(\frac{Wound\ area\ at\ 0h - Wound\ area\ at\ 24h}{Wound\ area\ at\ 0h}\right) \times 100$$

### 2.2.6. Statistical Analysis

GraphPad Prism version 9 software was used to perform statistical analysis. One-Way ANOVA Kruskal–Wallis test was performed to determine the significant differences in cell viability and evaluate apoptotic rates. Mann–Whitney test was used to evaluate the wound-healing assay. *p* values < 0.05 were considered statistically significant.

## 3. Results

### 3.1. The Anti-Proliferative Effect of Avermectin B1a on HCT-116

The anti-proliferative activity of avermectin B1a against HCT-116 colon cancer cells was evaluated using the MTT assay. The results showed that avermectin B1a reduced HCT-116 cell viability and displayed good anti-proliferative activity in a dose-dependent manner, with an IC$_{50}$ value of 30 μM. The highest inhibitory effect on the cell survival rate was observed after 24 h. Therefore, in our experiments, we used 30 μM of avermectin B1a, as this concentration had the greatest and statistically significant inhibitory effect on the cell survival rate in comparison to the other tested concentrations of the compound at various time intervals.

Furthermore, as shown in Figure 1, cell viability decreased to 50.1%, and the percentage of viable cells was 49.9% after 24 h of treatment with 30 μM of avermectin B1a (*p* = 0.006 **); therefore, the 30 μM concentration was considered as the half maximal inhibitory

concentration ($IC_{50}$). Additionally, our data showed that avermectin B1a did not significantly influence cell viability at concentrations of 2.5, 5, 10, 15, and 20 µM. As a result, our findings indicate that avermectin B1a is most potent against HCT-116 cells at a concentration of 30 µM; consequently, this concentration was selected for use in further experiments.

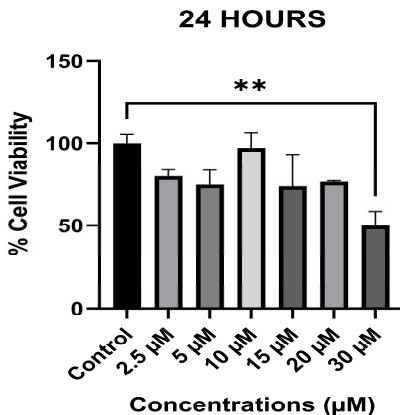

**Figure 1.** Viability of HCT-116 cells in the presence of avermectin B1a. HCT-116 colon cancer cells were treated with different concentrations of avermectin B1a for 24 h. Viability was determined using an MTT assay. The non-treated cells were considered as having 100% viability. $p < 0.01$ ** was considered statistically significant compared to the control.

### 3.2. The Effect of Avermectin B1a on Tubulin Polymerization

We investigated whether avermectin B1a modulates microtubule dynamics since ivermectin, the analog molecule of avermectin B1a, stabilized mammalian microtubules by increasing polymerization and has been identified as a stabilizer of tubulin polymerization [20]. Optical density is used to track the development of polymerized microtubules over time in the in vitro tubulin polymerization experiment, which is the standard [26]. Since purified recombinant $\alpha$- and $\beta$-tubulins do not polymerize efficiently, highly purified cellular tubulin preparations that include microtubule-associating proteins are typically utilized to enable efficient polymerization kinetics [27].

To explore the tubulin polymerization triggered by avermectin B1a, >97% pure tubulin was treated with avermectin B1a and control compounds for 1 h at 37 °C. The proportion of polymerized microtubule components was tracked by measuring the absorbance at 340 nm at regular intervals of 30 s for an hour. It was observed that the compound avermectin B1a had a substantial influence on microtubule polymerization at 30 µM; this influence consisted of increasing the polymerization of mammalian tubulin, which might alter the dynamics of tubulin polymerization and depolymerization and may lead to cell death. Meanwhile, paclitaxel, a well-known microtubule stabilizer, was used to enhance protofilament assembly. As demonstrated in Figure 2, avermectin B1a, like paclitaxel, can stabilize tubulin assembly and might stimulate protofilament assembly, indicating that a potential microtubule stabilizer may be established. Paclitaxel had the highest stimulating activity among the investigated groups, while colchicine was the least active substance in this assay.

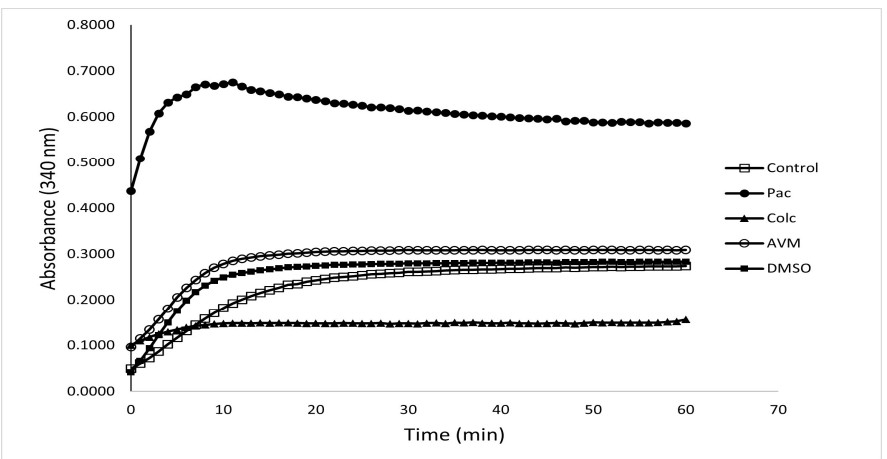

**Figure 2.** In vitro tubulin polymerization assay. The progress of tubulin polymerization in the presence of 30 µM of avermectin B1a, 10 µM of paclitaxel as a positive polymerization control, 10 µM of colchicine as a negative polymerization control and DMSO was investigated. The depicted increase in absorbance demonstrates the gradual polymerization of tubulin, which was promoted by avermectin B1a.

### 3.3. The Effect of Avermectin B1a on Apoptosis

The stimulation or inhibition of microtubule polymerization may result in the interruption of mitosis and eventually cell apoptosis, as many studies have shown that anti-mitotic cancer agents can also cause cell death by triggering apoptosis [28,29]. Therefore, it is important to investigate the apoptotic behavior of HCT-116 cells induced by avermectin B1a, since it has exhibited significant anti-proliferative activity and the disruption of microtubule assembly.

To evaluate the impact of avermectin B1a on apoptosis, an Annexin V-FITC/propidium iodide (PI) assay was conducted. The results demonstrated that apoptosis was induced (Figure 3). After 24 h of treatment with 30 µM of avermectin B1a, the percentage of cells undergoing apoptosis was found to be 39.83%, while this value was 18.07% in the control group. Furthermore, it was demonstrated that the apoptotic rate induced by avermectin B1a was significantly higher than that induced by DMSO, which was only 19.13%.

Overall, the presence of apoptotic and necrotic cells can be attributed to the fact that, "unlike" the DMSO group, the HCT-116 cells were exposed to a concentration of 30 µM of avermectin B1a for 24 h. Compared to the untreated cells, the results showed that avermectin B1a significantly promoted apoptosis in the HCT-116 cells (Figure 4) ($p < 0.0225$ *).

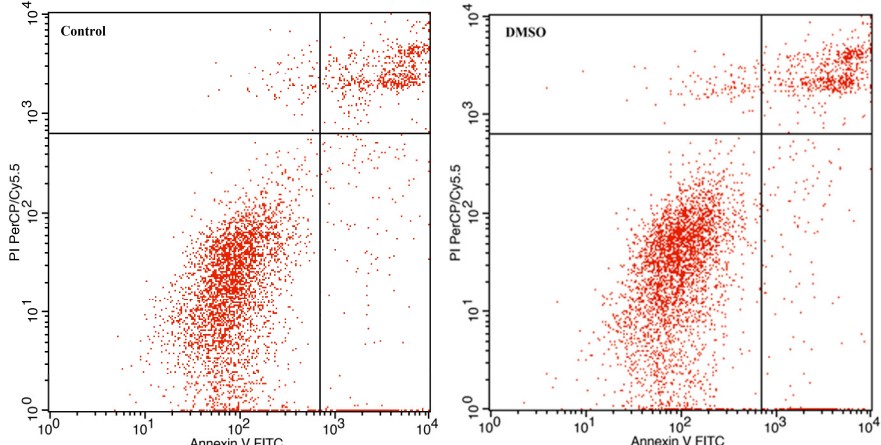

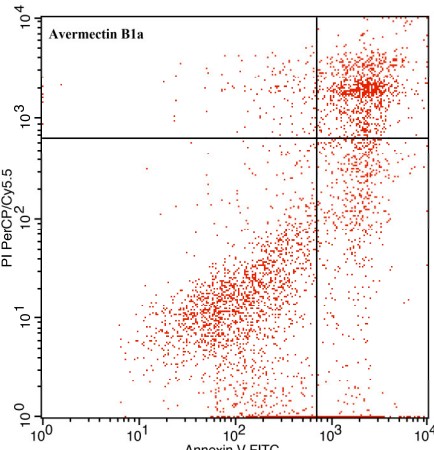

**Figure 3.** Apoptotic effect of avermectin B1a on HCT-116 cells. The lower left quadrants show live cells, the lower right quadrants show early apoptotic cells, the upper right quadrants show late apoptotic cells, and the upper left quadrants show necrotic cells.

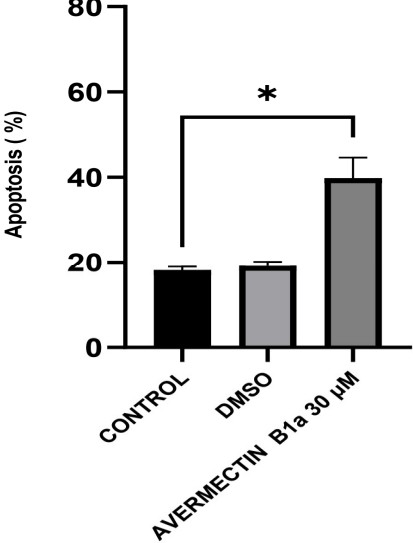

**Figure 4.** Percentage of apoptosis induced by avermectin B1a in HCT-116 cells. $p < 0.05$ * was considered statistically significant compared to the control.

### 3.4. The Effect of Avermectin B1a on Cell Migration

It has been validated that tubulin-targeting compounds have the ability to interfere with cell migration by blocking microtubule remodeling in the trailing region of the cell, where more active microtubules are usually found [30]. To study the effect of avermectin B1a on cell migration, the average migration rates of the scratched HCT-116 cells' monolayers were calculated. As shown in Figure 5, after 24 h of incubation, the cell migration inhibition percentage of HCT-116 cells in the control group was 98.08%, which can be compared to a value of 90.41% for DMSO, whereas it decreased to 15.61% following treatment with 30 μM of avermectin B1a. This result showed that the compound avermectin B1a may considerably reduce the migration of HCT-116 cells, which could be a starting point for the discovery of new microtubule-stabilizing agents.

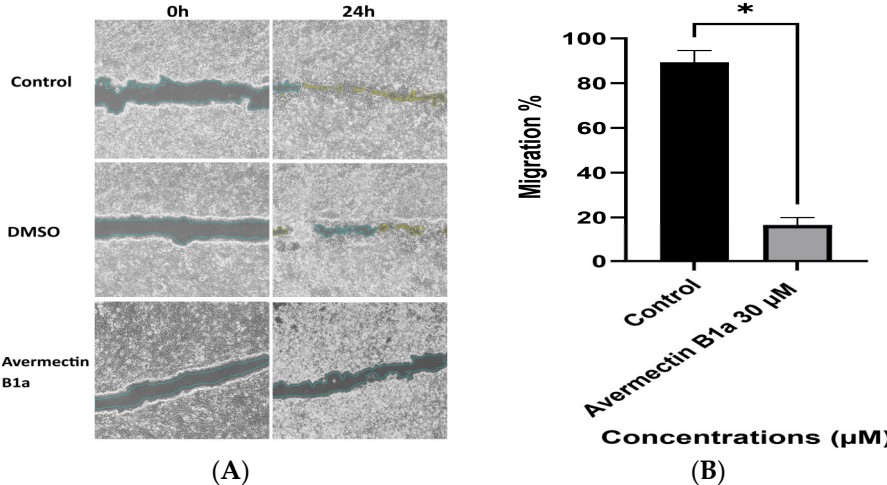

**Figure 5.** (**A**) Representative images from in vitro wound-healing assay. HCT-116 cells were treated with 30 μM avermectin B1a and DMSO for 24 h. (**B**) The summary bar graph represents the percentage of cell migration inhibition after 24 h with treatment. Values are means ± Standard Error (SE) of four replications (p ≤ 0.05 *).

## 4. Discussion

In this study, a compound with a macrocyclic lactone structure was investigated in an attempt to develop a potential anti-cancer drug from natural derivative small molecules. Avermectin B1a is a commonly utilized chemical in the avermectin family that is effective for both agricultural and medicinal uses [6]. Previous research has shown that avermectin family compounds have significant anti-cancer properties and are more effective at inhibiting tumor cell development [31–34].

The compound avermectin B1a, with the ability to interact with tubulin, has antiproliferative and anti-migration properties and induces apoptosis in colon cancer cells. In the current study, we tested the effect of avermectin B1a on cell proliferation and found that avermectin B1a inhibited the growth of HCT-116 cancer cells with $IC_{50}$ value at the micromolar level (Figure 1). This suggests that avermectin B1a may be helpful in the antiproliferative therapy of cancer and other diseases.

In developed countries, colon cancer is one of the most prevalent human cancers [35]. Numerous treatments for colon cancer have been discovered and tested in preclinical and clinical trials [36]. Since proteins are involved in the mechanism of cancer, chemicals that can interact with autophagy, cell cycle, apoptosis, and/ or necrosis-related proteins may result in significant structural and functional alterations which can be a potential treatment in cancer therapy [37]. Proteins have indeed been interpreted to be a very significant class of biomacromolecules with unique structures, dynamics, and functions, including those in the cytoskeleton, like tubulin [38], which has recently received significant attention as a promising therapeutic target for cancer [13,39]. In this study, avermectin B1a exhibited tubulin-stabilizing activity at a specific concentration. The results show that avermectin B1a has higher activity compared to DMSO and colchicine and exhibits a paclitaxel-like effect. The findings of our study can potentially provide valuable information with respect to creating anti-cancer medicines, particularly those targeting tubulin polymers.

To the best of our knowledge, apoptosis has been well recognized as a distinct pathologic process in colon cancer, and apoptosis inducers have been used to treat colon cancer [40]. Regarding colon cancer, the apoptotic effects and anti-cancer mechanisms of avermectin B1a remained unknown. The apoptotic properties of this compound that affect HCT-116 colon cancer cells were investigated in this study. The flow cytometry results demonstrated that the apoptotic rate of avermectin B1a at a 30 μM concentration was

39.83% after 24 h of treatment. According to our findings, the cells treated with our compound of interest exhibit a higher rate of apoptosis, suggesting that avermectin B1a induces apoptosis in cancer cells.

Cell migration is a key cause of mortality and morbidity among individuals with colon cancer and drives cell invasion and metastatic progression [41]. We performed wound-healing and cell migration tests to determine the possible effects of avermectin B1a on HCT-116 cells. According to the results of the wound-healing assay, avermectin B1a significantly reduced wound healing at a dose of 30 µM. After 24 h of treatment, the HCT-116 cells' migration rate was 15.61%. Furthermore, after 24 h, the cell migration rates of the HCT-116 cells treated with DMSO and the untreated cells were 90.41% and 98.08%, respectively. Thus, at a given dose and duration, avermectin B1a could prevent the migration of HCT-116 cells.

### 5. Conclusions

Avermectin B1a, a member of the avermectin family and a naturally occurring molecule, was examined in a colon cancer cell line in order to identify novel microtubule-targeting agents for cancer therapy. An anti-proliferative assay revealed that compound avermectin B1a showed anti-proliferative activity against the HCT-116 cancer line, presenting an $IC_{50}$ value of 30 µM. According to the study's results, avermectin B1a may be able to accelerate tubulin protofilament assembly by imitating the effects of paclitaxel; this would increase the rate at which tubulin polymerizes. Avermectin B1a may also increase apoptosis in HCT-116 cells and decrease the rate of cell migration. In light of the data presented here, this naturally derived molecule might be worthy of consideration as a possible leading microtubule-targeting agent for the development of future anti-cancer drugs.

**Author Contributions:** Conceptualization, Q.H., D.G.R., and O.D.; methodology, D.G.R.; statistical analysis, Q.H., D.G.R., and O.D.; investigation, Q.H.; writing—original draft preparation, D.G.R., and O.D.; writing—review and editing, O.D.; supervision, O.D.; project administration, Q.H.; funding acquisition, Q.H. All authors have read and agreed to the published version of the manuscript.

**Funding:** This research received no external funding.

**Institutional Review Board Statement:** Not applicable.

**Informed Consent Statement:** Not applicable.

**Data Availability Statement:** Not applicable.

**Conflicts of Interest:** The authors declare no conflicts of interest.

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
