# Peer review of "Avermectin B1a Shows Potential Anti-Proliferative and Anticancer Effects in HCT-116 Cells via Enhancing the Stability of Microtubules"

_cimb, doi:10.3390/cimb45080395_

Round 1

Reviewer 1 Report

­­The manuscript shows preliminary evidences ­­­­­­­that Avermectin B1a has possible anti-cancer effects. The only major comment in my opinion is that the evidences presented in the manuscript are based on only one cell line used, HCT-116. The experiments conducted on HCT-116 alone is not be enough to attest the compound’s anti-proliferative effects on colon cancer. This manuscript could use results at least with a few other cell lines.

The other minor edit I want to suggest is to improve the image quality of the figure1. It appears to be blurred in the current version.

Author Response

Regarding your major comment about the use of only one cell line, HCT-116, in our study, we acknowledge its limitation. We understand the importance of validating our findings across multiple cell lines to establish the broader applicability of Avermectin B1a's potential anti-proliferative effects on colon cancer. Unfortunately, due to very limited funding, we were unable to purchase additional cell lines for this particular study, but we plan to design another project to expand our investigation to include multiple cell lines to strengthen the validity and generalizability of our findings.

Furthermore, we appreciate your minor edit suggestion to improve the image quality of Figure 1. We apologize for the blurriness in the current version. We carefully revised and enhanced the image quality to ensure clarity and readability.

Reviewer 2 Report

The paper starts with the wrong assumption that microtubule polymerization can be an indication of anti cancer activity.

Anticancer drugs either stabilize (block polymerization) or destibilize microtubules (as reported also in reference 15 of the paper).

Anyway, from fig. 2, it appears that microtubule behaviour with AVM is like that with colchicine, meaning that microtubules dynamics is blocked. Therefore, I suggest changing the title of the paper, saying that AVM blocks microtubule polymerization. In this contest, I suggest better explain the fig legend.

Author Response

In our study, we aimed to investigate the effect of Avermectin B1a on tubulin polymerization. To assess this, we employed two controls to validate our findings effectively. We utilized paclitaxel as a positive control in our experiments, which is known as a Tubulin stabilizing agent and a widely recognized anticancer drug. Paclitaxel is well-known for its ability to enhance tubulin polymerization. By comparing the effects of Avermectin B1a with those of paclitaxel, we could establish whether Avermectin B1a exhibited similar properties in promoting tubulin polymerization.

As a negative control, we employed colchicine, which is known as a Tubulin destabilizing agent. In an other word, colchicine is recognized as a tubulin polymerization inhibitor. This negative control allowed us to confirm that our results were specific to the actions of Avermectin B1a and not influenced by nonspecific factors.

As a result of our experimental data, Avermectin B1a demonstrated a higher absorbance than the negative control (colchicine) and closely resembled the positive control (paclitaxel) in enhancing the stability of microtubules. These observations indicate that Avermectin B1a indeed promotes tubulin polymerization, similar to paclitaxel.
